# The Association between Blood Indexes and Immune Cell Concentrations in the Primary Tumor Microenvironment Predicting Survival of Immunotherapy in Gastric Cancer

**DOI:** 10.3390/cancers14153608

**Published:** 2022-07-25

**Authors:** Jiajia Yuan, Xingwang Zhao, Yanyan Li, Qian Yao, Lei Jiang, Xujiao Feng, Lin Shen, Yilin Li, Yang Chen

**Affiliations:** 1Department of Gastrointestinal Oncology, Key Laboratory of Carcinogenesis and Translational Research (Ministry of Education), Peking University Cancer Hospital and Institute, Beijing 100142, China; jyuan@bjmu.edu.cn (J.Y.); zhaoxingwang237@163.com (X.Z.); qiuyesiyu3@163.com (Y.L.); ncuskjianglei@163.com (L.J.); 13293656798@163.com (X.F.); shenlin@bjmu.edu.cn (L.S.); 2Department of Pathology, Key Laboratory of Carcinogenesis and Translational Research (Ministry of Education), Peking University Cancer Hospital and Institute, Beijing 100142, China; yaoqian@bjmu.edu.cn

**Keywords:** gastric cancer, immunotherapy, PD-1/PD-L1, tumor microenvironment, blood index

## Abstract

**Simple Summary:**

Due to the complexity of the immune response, no single biomarker is available for adequate patient stratification in the context of gastric cancer (GC). In this study, we used multiplexed immunohistochemistry combined with digital image analysis to uncover the immune cell features in 80 patients with GC. Furthermore, we analyzed the association of blood indexes with the primary gastric cancer immune microenvironment. Then, we validated the predicted value of the blood index in a larger GC cohort (*n* = 357) receiving anti-PD-1/PD-L1 immunotherapy. Importantly, this approach allowed us to map rare cell types with complex phenotypes, characterize the PD-1 and PD-L1 expression intensity in situ, and assess the biomarker value of these parameters and their associations with the blood index. Our data suggest that blood indexes, associated with primary tumor microenvironment, can be used to predict the immune related prognosis in GC.

**Abstract:**

The tumor microenvironment plays a vital role in tumor progression and treatment response. However, the association between immune cell concentrations in primary tumor and blood indexes remains unknown. Thus, we enrolled patients with gastric cancer (GC) in two cohorts. We used multiplexed immunohistochemistry to quantify in situ proteins covering rare cell types at sub-cellular resolution in 80 patients with GC in the first cohort. A high correlation between the LMR (lymphocyte-to-monocyte ratio)/NLR (neutrophil-to-lymphocyte ratio) and tumor immune microenvironment was found. The density of exhausted CD8 T cells including CD8^+^PD1^−^TIM3^+^, CD8^+^LAG3^+^PD1^+^, CD8^+^LAG3^+^PD1^−^, CD8^+^LAG3^+^PD1^+^TIM3^−^ was negatively associated with LMR and positively associated with NLR (*p* < 0.05). Additionally, the higher density of macrophages in tumor core was associated with a higher platelet-to-lymphocyte ratio and systemic immune-inflammation index. Furthermore, we validated the prognostic value of LMR and NLR in an independent cohort of 357 gastric cancer patients receiving immunotherapy. Higher LMR at baseline was significantly associated with superior immune-related PFS (irPFS) and a trend of superior immune-related OS (irOS). Higher NLR was associated with inferior irOS. In conclusion, blood indexes were associated with immune cells infiltrating in primary tumors of GC. NLR and LMR are associated with the density of exhausted CD8^+^ T immune cells, which leads to prognostic values of immunotherapy.

## 1. Introduction

Gastric cancer (GC), with high incidence and death rates in China and worldwide, remains a major threat to public health [1]. Immunotherapy has become a standard for first-line and later-line therapy of GC [2]. CheckMate 649, a phase 3 clinical trial comparing immunotherapy plus chemotherapy versus chemotherapy alone, suggested nivolumab plus chemotherapy resulted in significant improvements in OS, with PD-L1 CPS of one or more and PFS with PD-L1 CPS of five or more [3]. However, the response rate for the general population of GC patients with immunotherapy is 60%, 15% higher than chemotherapy alone, which is far from enough. Although the PD-L1 CPS score suggested a higher score may give a high response rate, other biomarkers for response prediction are urgently needed [4]. 

The tumor microenvironment, including tumor immune cells such as T cells, B cells, neutrophils and macrophages, and their precise location in relation to cancer cells, may influence the therapeutic response [5]. Immunohistochemistry has been widely used to analyze immune cells in patients with GC, such as CD3, CD8, PD-1, PD-L1, CD163, and LAG3 expression [6,7,8,9]. However, traditional immunohistochemistry only stained one marker in one slide and could not identify rare cell types with the combination of several markers. Thus, multiplexed immunohistochemistry (m-IHC), which can simultaneously detect multiple antigens in situ with single cell resolution, has been developed [10]. 

Studies from our team have suggested that blood indexes including CD4^+^ T lymphocytes are related to the response to immunotherapy in gastrointestinal cancer [11]. Moreover, the lymphocyte-to-monocyte ratio (LMR) serves as an independent prognostic factor for GC patients treated with ICIs [12]. However, the relationships of blood index and the primary tumor microenvironment were largely unknown. Hence, in order to elucidate the relationship between immune cell concentrations and verify the prognostic factors of blood index, we studied two independent retrospective cohorts of GC. In the first cohort, we examined the relationship between immune cells in primary tumor and blood indexes. In the second cohort, we validated the prognostic value of blood index in GC patients receiving immunotherapy.

## 2. Materials and Methods

### 2.1. Study Population

We performed a tumor immune microenvironment evaluation of 80 GC patients enrolled at Peking University Cancer Hospital and Institute from July 2014 to December 2019, comprising the first cohort. An independent validation cohort included 357 GC patients receiving anti-PD-1/PD-L1 based therapy between November 2016 and May 2021 at Peking University Cancer Hospital was established. Written informed consent was signed by the patients or their legal guardians before immunotherapy. All blood tests and treatments were performed in accordance with institutional guidelines. Clinical data collected from patients’ electronic medical records included demographic information, histology, and laboratory tests results. The inclusion criteria for the validation cohort were: (1) pathologically confirmed GC; (2) administration of anti-PD-1/PD-L1-based treatment regimens. The exclusion criteria were: (1) incomplete hematological data; (2) lost follow-up.

Patients were observed until death or end of follow-up (5 June 2021), whichever came first. Dates of death were obtained through telephone call-based follow-up by doctors or the follow-up center in our hospital. The study protocol was approved by the Ethics Committee of the Peking University Cancer Hospital and Institute.

### 2.2. Assessment of Tumor Microenvironment

Multiplexed immunohistochemistry (m-IHC) staining was performed to visualize the expression of CD8, PD-1, TIM3, LAG3, CD4, FoxP3, CTLA4, PD-L1, CD68, CD163, HLADR, STING, CD20, CD66b, CLDN18.2 and CD147. Tissues, within 30 min after being excised, were fixed in formalin for 24–48 h, dehydrated and embedded in paraffin, then cut into sections with a thickness of 4 μm. FFPE slides were melted and dehydrated at 60 °C for 12 h, deparaffinized and rehydrated using xylene and alcohol, and then placed in citrate buffer (pH 6.0) for FoxP3 staining or EDTA buffer (pH 9.0) for others, and the whole reactive system was placed in a microwave oven for heat-induced antigen retrieval. Then, the sections were blocked by a commercially available blocking buffer (X0909; Dako, Santa Clara, CA, USA) for 10 min. Appendix A shows the antibodies used for the staining. The slides were incubated with the primary antibody and horseradish peroxidase-conjugated secondary antibody, and tyramine signal amplification (TSA) was performed in accordance with the pre-optimized antibody concentration and the order of staining. Antibody stripping and antigen retrieval were performed after each round of TSA. 4′, 6-diamidino-2-phenylindole (DAPI, Sigma-Aldrich, St. Louis, MO, USA; cat. D9542) was used to stain nuclei. All stained GC specimens were evaluated by two experienced pathologists to ensure that they met the requirements for further analysis.

Images were acquired using the Mantra Quantitative Pathology Imaging System (PerkinElmer, Waltham, MA, USA). A whole slide scan of the multiplex tissue sections produced multispectral fluorescent images visualized in Phenochart (PerkinElmer, Waltham, MA, USA) software. Representative regions of interest (ROI) were chosen by a specialized pathologist, and multiple fields of view were acquired at 20× power for further analysis. The multispectral images were analyzed using InForm image analysis software 2.4 (PerkinElmer, Waltham, MA, USA). Density of cells in each ROI was calculated by combining the cell counts from all images and normalizing by the total area (cell/mm^2^).

### 2.3. Assessment of Hematological Parameters

Blood samples were routinely collected prior to therapy (Day 0 or 1) and every 7 days. Blood indexes included lymphocytes (L), monocytes (M), platelets (P), and neutrophils (N). We also calculated neutrophil-to-lymphocyte ratio (NLR), lymphocyte-to-monocyte ratio (LMR), platelet-to-lymphocyte ratio (PLR), and systemic immune-inflammation index (SII) at the start of immunotherapy. SII was defined as P× N/L. Immune related overall survival (irOS) was defined as the time from initial anti-PD-1/PD-L1 based treatment to death. Immune related progression-free survival (irPFS) was defined as the time from initial anti-PD-1/PD-L1 based treatment to disease progress or death. Censoring occurred if patients were still alive at last follow-up.

### 2.4. Statistical Analysis

Our primary hypothesis was based on the assessment of an association of blood indexes including NLR, LMR, PLR, and SII at baseline with the immune cell infiltration in primary tumor location. Our second aim was to analyze the blood index-related mortality in a multivariable-adjusted Cox proportional hazards regression model. We initially included the variables of age (<60 vs. ≥60), sex (male vs. female), tumor location (GEJ vs. Non-GJE), Lauren classification (intestinal type vs. diffused type vs. mixed type), lines of therapy (1 vs. 2 vs. ≥3), and types of therapy (immunotherapy alone vs. combination with other therapy). We conducted a backward elimination with a threshold of *p* = 0.05 to select variables for the final models. For cases with missing information in any of the categorical covariates (tumor location (0.92%), tumor differentiation (7.34%), Lauren classification (6.42%), HER2 expression (9.17%), MMR status (10.09%), PD-L1 expression (11.93%), and EBER status (19.27%)), we included these cases in the majority category of a given covariate. We implemented the Kapan–Meier method to estimate the distribution of irPFS and irOS, and log-rank test into our analyses. All statistical analyses were performed using SPSS (Version 20, Chicago, IL, USA) and software package R (version 4.2.1). Packages of “survival” and “survminer” were used in R analysis. All *P* values were two-sided, and statistical significance was considered at *p* < 0.05.

## 3. Results

### 3.1. Clinicopathological Features of Gastric Cancer Patients

Eighty patients were enrolled in the first cohort for microenvironment analysis (Table 1). The median age of the patients was 60 years (range, 54–66 years), and the majority of patients were men (76.3%). The clinicopathological features of 357 patients in the validation cohort are also presented in Table 1. Among 357 patients, 45 (12.6%) were dMMR, and 26 (7.3%) were EBV positive, respectively.

### 3.2. Blood Indexes Were Associated with Immune Cell Concentrations in Primary Tumor

To investigate the tumor-infiltrating immune cells (TIICs) within GC, we quantified the densities of immune cells in 80 full-face formalin-fixed paraffin-embedded (FFPE) samples with m-IHC staining. Cell phenotyping data were obtained based on positivity and relative intensity of all markers in one panel. Cell population densities were calculated for “all” regions (tumor + stroma) and measured separately in tumor and stroma. TIICs were analyzed at a single-cell level, and 26 major populations were characterized (Figure 1). 

The density of exhausted CD8^+^ T cell was significantly positively correlated with NLR (*p* = 0.029), PLR (*p* = 0.007), and SII (*p* = 0.047). In detail, higher NLR was associated with higher CD8^+^LAG3^+^PD1^+^ (*p* = 0.015), higher CD8^+^LAG3^+^PD1^−^ (*p* = 0.011), higher CD8^+^LAG3^−^PD1^+^ (*p* = 0.039), higher CD8^+^PD1^−^TIM3^+^ (*p* = 0.010), and higher CD8^+^LAG3^+^PD1^+^TIM3^−^ (*p* = 0.005). Inversely, higher LMR was associated with lower CD8^+^LAG3^+^PD1^+^ (*p* = 0.045), lower CD8^+^LAG3^+^PD1^−^ (*p* = 0.023), lower CD8^+^PD1^−^TIM3^+^ (*p* = 0.031), and lower CD8^+^LAG3^+^PD1^+^TIM3^−^ (*p* = 0.012). Furthermore, PLR was associated with higher CD8^+^LAG3^+^PD1^−^ (*p* = 0.022), higher CD8^+^LAG3^−^PD1^+^ (*p* = 0.016), higher CD8^+^LAG3^−^PD1^−^ (*p* = 0.014), higher CD8^+^PD1^−^TIM3^+^ (*p* = 0.004), and higher CD8^+^LAG3^−^PD1^+^TIM3^−^ (*p* = 0.026). SII was associated with higher CD8^+^LAG3^+^PD1^−^ (*p* = 0.029), higher CD8^+^LAG3^−^PD1^−^ (*p* = 0.046), and higher CD8^+^PD1^−^TIM3^+^ (*p* = 0.012). In addition, the density of CD4^+^ T cells only showed relevance with PLR (*p* = 0.041), especially in the CD4^+^FOXP3^−^ group (*p* = 0.027), whereas the density of CD68^+^ macrophages in tumor score was associated with PLR (*p* = 0.004), and SII (*p* = 0.010), especially in the CD68^+^CD163^−^ macrophage group (*p* = 0.001 and *p* = 0.002, separately), and STING-altered CD68^+^ macrophages were associated with SII (*p* = 0.010) (Figure 2, Appendix A). We observed the same trends in the tumor core, tumor region and stromal regions (Appendix A).

### 3.3. Blood Indexes Predicting Survival in GC Patients Receiving Anti-PD-1/PD-L1 Immunotherapy

We retrospectively included 357 GC patients with available data on baseline blood indexes including neutrophils, monocytes and lymphocytes who received anti-PD-1/PD-L1-based treatments (1st, or 2nd, or 3rd line immunotherapy). We calculated LMR, NLR, PLR, and SII at baseline. Median irPFS and irOS after therapy initiation were 6.6 (95%CI: 5.3–8.4) and 16.3 (95%CI: 13.0–18.8) months, respectively. 

We examined the association of NLR, LMR, PLR, and SII at baseline with survival (Table 2 and Table 3). In univariate analysis, higher NLR was significantly associated with inferior irOS (*p* = 0.013, HR = 1.40, 95%CI: 1.10–1.90). Additionally, higher SII was associated with a trend of inferior irOS (*p* = 0.077, HR = 1.30, 95%CI: 0.97–1.70), and higher LMR with a trend of superior irOS (*p* = 0.085, HR = 0.78, 95%CI: 0.59–1.00).

Furthermore, LMR from baseline was an independent prognostic factor for irPFS in multivariate analysis (*p* = 0.034, adjusted HR = 0.66, 95%CI: 0.45–0.97), and monocytes from baseline were an independent prognostic factor for irPFS in both univariate (*p* = 0.015, HR = 1.50, 95%CI: 1.10–2.00) and multivariate analysis (*p* = 0.003, adjusted HR = 1.82, 95%CI: 1.22–2.71). Figure 3 and Appendix A show Kaplan–Meier curves for irPFS and irOS according to blood indexes.

## 4. Discussion

We present a detailed multistep platform for multispectral imaging of tissues that generates high-quality datasets at single-cell resolution for biomarker discovery and quantitative pathology to characterize the tumor immune microenvironment and to guide precision immunotherapy in GC. To our knowledge, this is the first study to investigate the association of blood indexes with rare immune cell types including CD8 exhausted T cells in the primary tumor core. In addition, this is also the largest validation cohort of GC receiving anti-PD-1/PD-L1 therapy.

CD8 T cell subsets inform the mechanism of immune checkpoint inhibitors. We adopted a detailed panel of markers providing potential biology of T cells. Conflicting prognostic roles of CD8 T cells have been reported and may be due to patients with higher CD8 T cell densities also having higher PD-L1 expression [13]. In our work, we classified CD8 T cells into precise categories. We found that almost all CD8 T cell categories, regardless of LAG3, PD-1, or TIM3 expression, were associated with NLR, indicating that NLR could act as a specific marker for the density of CD8 T cells in primary tumor. Moreover, we found that PLR was more likely associated with CD8^+^PD1^−^ and CD8^+^LAG3^−^ subsets, suggesting a potential benefit for the therapeutic population. In addition, M1 macrophages were suggested to be associated with PLR and SII, which may help distinguish immune tolerance and resistance.

Previous reports suggested that CD8 T cells could be induced to gradual deterioration of T cell function by excessive amounts of signals, a state called “exhaustion” [14]. Exhausted T cells express multiple inhibitory receptors, such as PD-1, LAG3, and TIM3 [15]. Targeting PD-1 together with LAG3 facilitates T cell reinvigoration [16]. Our study found a subset of pre-exhausted T cells that could be distinguished at baseline, suggesting single anti-PD-1 therapy might be effective in initial treatment [17]. Additionally, monitoring blood indexes would let us know when to add combination therapy as exhausted cells appear and proliferate [18]. Hsu and colleagues found that exhausted CD8^+^LAG3^+^ T cells could predict efficacy of single-agent immunotherapy versus immunotherapy combination therapies in hepatocellular carcinoma. Due to the small population, no significant difference was achieved, but the trend preferred combination immunotherapies [19]. Large-scale evidence is needed.

This study offers new insights into the nature of GC immunity and how this information can be harnessed towards effective immunotherapy strategies. From our study, we learned that GC cells have a population that can benefit from immunotherapy using anti-PD-1 alone, suggesting a potential reason for CheckMate649 to succeed with 60% ORR [3]. It will be interesting and cost-effective to see a variety of patterns of blood indexes before treatment, which can predict different responses. Combination of immunotherapy, such as antibody against PD-1/PD-L1 with antibody against LAG3, TIM3, or CTLA4, could further increase tumor-infiltrating lymphocyte functions [20]. This provides a way to reverse the remaining 40% of primary resistance and large amount of acquired resistance in GC.

Our study has several limitations. Immune cell subsets can be related to one another, as “daughter cells” with distinct phenotypes are derived from the same “ancestor cell”. Thus, independent studies are warranted to confirm our findings. The strengths of the study include the acquisition of the entire TME in the whole slides, followed by standardized selection of discrete ROIs. These findings highlight the benefit of using blood indexes in the context of their relation with primary tumor. While our results support the prognostic significance of blood indexes in gastric cancer, further investigations are required to confirm the findings.

## 5. Conclusions

Our results support blood indexes as a robust, quantitative prognostic and predictive tool in gastric cancer. The exploration of the microenvironment composition of gastric cancer samples would offer critical insights into the complex and heterogeneous immune landscape associated with blood indexes and immunotherapy survival.

## Figures and Tables

**Figure 1 cancers-14-03608-f001:**
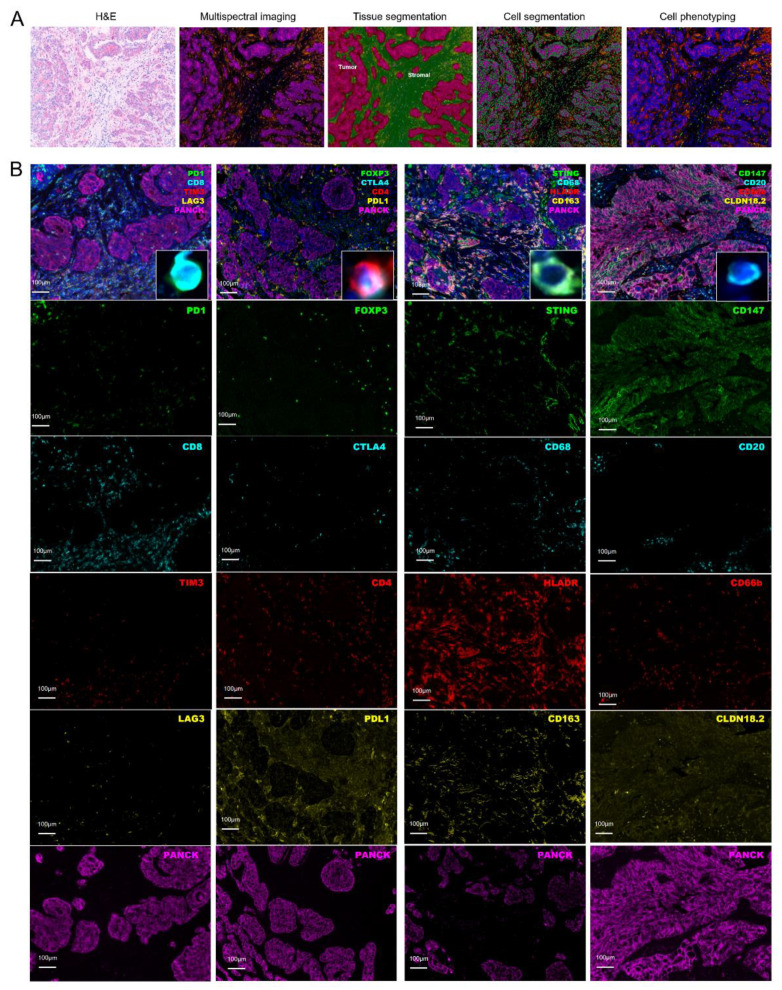
(**A**) Procedure for quantifying the densities of immune cells in FFPE samples with m-IHC staining. (**B**) Examples of m-IHC staining to visualize the expression of CD8, PD-1, TIM3, LAG3, CD4, FoxP3, CTLA4, PD-L1, CD68, CD163, HLADR, STING, CD20, CD66b, CD147, CLDN18.2 in primary gastric cancer microenvironment.

**Figure 2 cancers-14-03608-f002:**
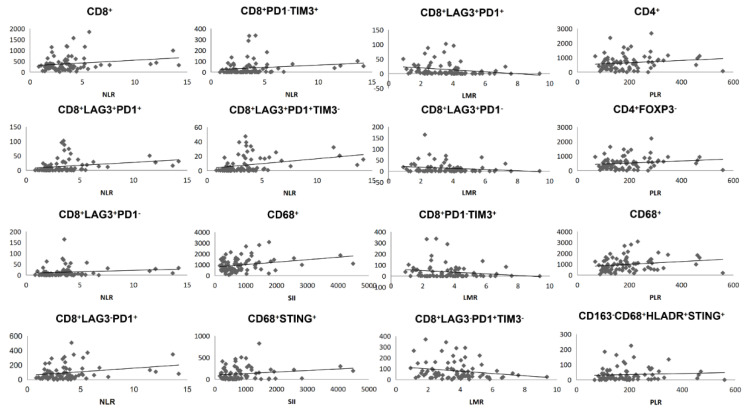
The relationship of NLR, LMR, PLR, and SII with the density of tumor infiltrating immune cells in tumor core.

**Figure 3 cancers-14-03608-f003:**
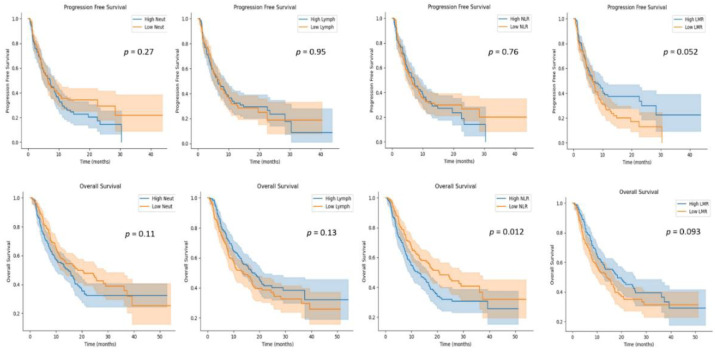
Kaplan–Meier curves for PFS and OS according to blood indexes.

**Table 1 cancers-14-03608-t001:** Baseline characteristics of gastric cancer patients.

Characteristic *	Total (Test Cohort) *n* = 80	Total (Validation Cohort) *n* = 357
Age		
Median, IQR	60 (54–66)	59 (51–65)
Sex		
Male	61 (76.3%)	255 (71.4%)
Female	19 (23.7%)	102 (28.6%)
ECOG PS		
0	49 (61.3%)	166 (46.5%)
≥1	31 (38.7%)	191 (53.5%)
Location		
GEJ	24 (30.0%)	112 (31.4%)
Non-GEJ	56 (70.0%)	245 (68.6%)
Differentiation		
High	0 (0%)	1 (0.3%)
Moderate	23 (28.8%)	88 (24.6%)
Moderate-poor	22 (27.5%)	62 (17.4%)
Poor	35 (43.7%)	195 (54.6%)
NA	0 (0%)	11 (3.1%)
Lauren classification		
Intestinal type	38 (47.5%)	138 (38.7%)
Diffused type	18 (22.5%)	89 (24.9%)
Mixed type	24 (30.0%)	79 (22.1%)
NA	0 (0%)	51 (14.3%)
Stage		
I	3 (3.8%)	1 (0.3%)
II	9 (11.3%)	4 (1.1%)
III	29 (36.2%)	27 (7.6%)
IV	39 (48.7%)	324 (90.7%)
NA	0 (0%)	1 (0.3%)
HER2 expression		
Positive	22 (27.5%)	75 (21.0%)
Negative	58 (72.5%)	268 (75.1%)
NA	0 (0%)	14 (3.9%)
PD-L1 expression (CPS)		
≥10	36 (45.0%)	88 (24.6%)
5-10	10 (12.5%)	31 (8.7%)
1-5	17 (21.25%)	16 (4.5%)
<1	17 (21.25%)	96 (26.9%)
NA	0 (0%)	126 (35.3%)
MMR status		
pMMR	69 (86.25%)	266 (74.5%)
dMMR	11 (13.75%)	45 (12.6%)
NA	0 (0%)	46 (12.9%)
EBV status		
Positive	10 (12.5%)	26 (7.3%)
Negative	70 (87.5%)	285 (79.8%)
NA	0 (0%)	46 (12.9%)

* Percentage indicates the proportion of patients with a specific clinical, pathologic, or molecular characteristic among all patients. Abbreviations: dMMR, deficient mismatch repair; pMMR, proficient mismatch repair.

**Table 2 cancers-14-03608-t002:** Multivariate analysis of independent risk factors for disease progression.

Variables	Progression-Free Survival		Progression-Free Survival	
	Hazard Ratio	*p*	Hazard Ratio	*p*
	Multivariate analysis with LMR		Multivariate analysis with Monocytes	
Lauren		0.013		0.0056
Intestinal	1		1	
Diffuse	1.65 (1.11–2.45)		1.77 (1.18–2.64)	
Peritoneal		0.69		0.84
Absent	1		1	
Present	1.09 (0.71–1.68)		0.96 (0.62–1.47)	
Line		0.026		0.0024
First Line	1		1	
Others	1.55 (1.05–2.28)		1.86 (1.25–2.78)	
LMR		0.034		
Low	1			
High	0.66 (0.45–0.97)			
Monocytes				0.0034
Low			1	
High			1.82 (1.22–2.71)	

**Table 3 cancers-14-03608-t003:** Multivariate analysis of NLR for death.

Variables	Univariate Analysis		Multivariate Analysis	
	Hazard Ratio	*p*	Hazard Ratio	*p*
Gender		0.003		0.28
Male	1		1	
Female	0.63 (0.47–0.86)		0.85 (0.57–1.29)	
Age		<0.001		0.0076
<60	1		1	
≥60	0.68 (0.51–0.9)		0.60 (0.41–0.88)	
Differentiation		0.023		0.25
Moderate	1		1	
Poor	1.4 (1–1.9)		0.62 (0.32–1.20)	
HER2		0.029		0.12
Negative	1		1	
Positive	0.65 (0.44–0.96)		0.61 (0.36–1.03)	
Peritoneal		0.00021		0.32
Absent	1		1	
Present	1.6 (1.3–2.1)		0.85 (0.57–1.28)	
Stage		0.0023		0.0091
I/II/III	1		1	
IV	3 (1.5–5.9)		3.41 (1.32–8.79)	
Line		<0.001		<0.001
First Line	1		1	
Others	2.6 (1.9–3.5)		2.60 (1.65–4.09)	
NLR		0.013		0.17
Low	1		1	
High	1.4 (1.1–1.9)		1.30 (0.90–1.89)	

## Data Availability

The datasets used and/or analyzed during the current study are available from the corresponding author on reasonable request.

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
