# Peer review of "The Association between Blood Indexes and Immune Cell Concentrations in the Primary Tumor Microenvironment Predicting Survival of Immunotherapy in Gastric Cancer"

_cancers, 2022, doi:10.3390/cancers14153608_

Round 1
Reviewer 1 Report
In this manuscript, the authors were trying to develop blood indexes as a marker to predict the immune-related prognosis in gastric cancer. The study is very important, but the manuscript is not well written since critical experimental details are lacking. Here are a few examples: the information on machine learning, and the information on the patients who received immunotherapy.
Table 2 is missing the manuscript text.
Author Response
Dear reviewer,
We appreciate your valuable comment. As you requested, we revised our manuscript from the following aspects:
First, we have added experimental details including how tissues were processed and how staining was done in the section "2.2 Assessment of tumor microenvironment." We also added Supplementary Table 1 including the information of primary antibody.
Second, please kindly find the details of patients received immunotherapy described in Table 1.
Third, we are sorry about the mistake of "missing Table 2 in the text". The number of Tables were re-ordered.
Fourth, the information on machine learning that we mentioned referred to the methods of automatically software. The inForm software actively learned the phenotyping algorithm from all spectrally unmixed images. Each DAPI-stained cell was individually identified according to its combination of fluorophore characteristics and cell morphology features associated with a segmented nucleus (DAPI signal). We revised the description in the manuscript.
Fifth, we asked a native speaker to double checked the English language and style.
We thank you for all the constructive comments. We hope you will find that the manuscript has been significantly revised and the concerns have been thoroughly addressed. Thank you again for your time and effort.
Reviewer 2 Report
The Abstract, Methods, Results and Discussion accurately described the content of this work. I consider the Introduction to be sound.
In order to improve results presentation as well as images visualization, I suggest enclosing insets in Figure 1, representing magnification of each image of immunohistochemistry.
Author Response
Dear reviewer,
We thank you for your positive comments and for your time and effort in reviewing our manuscript.
As you requested, we have enclosed insets in Figure 1. We also re-order and magnify each separated image, and hope it will be easier to read.
We also double checked English spell by a native speaker.
We thank you for the constructive comments. We hope you will find the concerns have been thoroughly addressed. Thank you again for your time and effort.
Round 2
Reviewer 1 Report
I am fine with authors' response.